# Assessment of Crystallinity Development during Fused Filament Fabrication through Fast Scanning Chip Calorimetry

**Dries Vaes [1], Margot Coppens [1], Bart Goderis [2]**  **, Wim Zoetelief [3] and Peter Van Puyvelde [1,\*]**

1   Department of Chemical Engineering, KU Leuven, Celestijnenlaan 200F box 2424, 3000 Leuven, Belgium
2   Department of Chemistry, KU Leuven, Celestijnenlaan 200F box 2404, 3000 Leuven, Belgium
3   DSM Materials Science Center, Urmonderbaan 22, 6167 RD Geleen, The Netherlands
\*   Correspondence: peter.vanpuyvelde@kuleuven.be

**Abstract:** Although semi-crystalline polymers are associated with considerably better mechanical properties and thermal stability compared to their amorphous counterparts, using them as feedstock for Fused Filament Fabrication still poses some major challenges. Hence, the development of printed part crystallinity during printing should be fully understood in order to control the developed weld strength, as well as part shrinkage and warpage. Infrared thermography was utilized to record the thermal history of deposited layers while printing a single-layer wall geometry, employing two PA 6/66 copolymers with distinct molecular weights as feedstock. Print settings were varied to establish which settings are essential to effectively monitor final part crystallinity. The resulting temperature profiles were simulated in a Fast Scanning Chip Calorimetry device that allows for the rapid heating and cooling rates experienced by the printed polymer. Both liquefier temperature and print speed were found to have very little influence on the total attained crystallinity. It became apparent that the build plate, set at a temperature above the polymer's glass transition temperature, imposes a considerable annealing effect on the printed part. A reduced molecular weight was observed to enhance crystallinity even more strongly. The presented methodology proves that Fast Scanning Chip Calorimetry is an effective tool to assess the impact of both print parameters and feedstock characteristics on the crystallization behavior of semi-crystalline polymers over the course of printing.

**Keywords:** additive manufacturing; fused filament fabrication (FFF); 3D printing; polymer crystallization; infrared thermography; fast scanning chip calorimetry

## 1. Introduction

The introduction of Additive Manufacturing (AM), more commonly referred to as 3D printing, in the late 1980s has made a significant impact on science and technology and revolutionized the way products are designed and manufactured [1,2]. As growth is expected to exceed 10 billion dollars for AM products and services by 2020, investments in AM related research and development by both industry and government have surged during the last decade [3,4]. AM comprises a group of manufacturing techniques all based on a similar working principle: fabricating three-dimensional (3D) parts by successive addition of material layers starting from computer-aided-design (CAD) models or 3D scans of existing objects [2,5]. Melt extrusion technologies, known as Fused Filament Fabrication (FFF) or Fused Deposition Modeling (FDM™), make up one of the largest groups amongst the polymer-based AM processes, which also include Selective Laser Sintering (SLS), stereolithography (SLA), and material jetting [1,6]. During FFF, feedstock in the form of a thermoplastic filament is fed by a pinch roller mechanism into a print head consisting of a liquefier, heated above the polymer melting

temperature, and a nozzle. The solid portion of the filament acts as a piston, exerting pressure to push the molten polymer out of the liquefier through the smaller print nozzle. While extruding molten polymer, the print head moves in the xy-plane following a prescribed pattern. After completing a layer, the heated build plate moves down one layer height, repeating the process to create a three-dimensional object layer-by-layer [5]. FFF has proven itself as a promising manufacturing technology, especially for low production volumes, allowing customization and freedom-of-design. However, its limited material palette, often inadequate mechanical properties and surface finishing remain major drawbacks [1,2].

Amorphous thermoplastic polymers, such as ABS, are extensively used in FFF. However, besides having good thermal stability, semi-crystalline thermoplastics often possess considerably better mechanical properties, making them tougher and more deformable compared to amorphous glassy polymers [7,8]. Hence, FFF with semi-crystalline polymers has found its use in, for example, biomedical applications, including PEEK [9] and PA [10] for the production of customized implants, and PLA [11] and PCL [12,13] for scaffolds. Nonetheless, semi-crystalline polymers exhibit far greater shrinkage upon cooling in comparison with amorphous plastics due to the considerable volume reduction associated with the formation of ordered, more densely packed regions during crystallization [14]. Crystallization of the extruded polymer melt upon deposition from the print nozzle will lead to anisotropic shrinkage and part warpage due to a build-up of internal stresses. This can severely disturb the dimensional stability of the printed part [15–17]. To provide the final printed part with adequate mechanical strength, polymer chains should diffuse and re-entangle across the interface between successively deposited layers to successfully weld them together [18]. This molecular interdiffusion can be appropriately described by the reptation model, especially for welding of amorphous polymer interfaces [19,20]. In the case of semi-crystalline interfaces, crystallization is believed to drastically lower molecular mobility, since polymer chains are confined within the formed crystalline structures [21–23]. Consequently, it is suggested that crystallization should take place after interlayer diffusion to establish sufficiently strong welds [24]. However, co-crystallization across the interlayer interface is considered to be an excellent reinforcement mechanism during welding of semi-crystalline interfaces [24,25].

Since both molecular diffusion and crystallization are strongly influenced by temperature, knowledge of the temperature history in a deposited layer is of key importance. Due to the subsequent deposition of molten polymer on top of a previously printed layer, the layer temperature will strongly change over time, making the FFF process highly non-isothermal. As a consequence, the development of interlayer strength and crystallization will become time-dependent [26,27]. Thermal monitoring of the FFF process to extract the non-isothermal temperature profile of a printed layer has been extensively reported in the literature, either by utilizing a thermocouple [28–30] or through infrared (IR) thermography [31–38]. Other authors have successfully attempted to model the temperature profiles within consecutively deposited layers [32,39–42].

The discussion above has illustrated the great significance of gaining insight in the behavior of semi-crystalline thermoplastics during FFF as a function of thermal history to possibly predict or tailor both the extent and gradient of crystallinity within a printed part and the resulting part quality in terms of mechanical strength and dimensional stability. Few studies, of which most are mainly focused on PLA, have investigated the influence of material properties or processing parameters, such as print speed and liquefier temperature, on the degree of crystallinity attained through printing. By using conventional Differential Scanning Calorimetry (DSC) and X-ray Diffraction (XRD), the crystallinity in the final printed part is examined showing little impact of the liquefier temperature, yet a large effect of increasing the build plate temperature. Annealing the printed part, either by the build plate heated sufficiently above the glass transition temperature ($T_g$) or as a follow-up treatment, has been proven to enhance crystallinity strongly [43–47]. The continuous successive deposition of new molten polymer onto previously printed layers results in consecutive annealing cycles, ultimately producing a gradient of crystallinity in the printed part with upper layers, having been exposed to fewer annealing cycles, being less crystalline. A slower print speed is also suggested to contribute to a higher degree of

crystallinity. For PLA, both a high L-enantiomeric purity and low molecular weight have been shown to greatly intensify crystallization kinetics, indicating the possibility to alter crystallization during FFF on a molecular level [30].

The presented research aimed at studying the effect of processing parameters, including print speed as well as liquefier and build plate temperature, on the thermal history and the resulting degree of crystallinity at different heights above the build plate for two PA 6/66 random copolymers with distinct molecular weights. By monitoring the layer temperatures during printing through IR thermography, temperature profiles as a function of time were obtained, which were then simulated in a Fast Scanning Chip Calorimetry (FSC) device. FSC has proven itself to be able to mimic the fast heating and cooling rates, which are typically experienced by the extruded polymer during the FFF process and cannot be achieved with conventional DSC [48]. Hence, by replicating the thermal conditions in the FSC device, the progression of crystallinity in a layer during printing and the contribution of each print setting can be monitored, once the layer temperature profile is known.

## 2. Materials and Methods

### 2.1. Printing of Wall Geometry

Two PA 6/66 random copolymers in the form of filament spools were used for printing as well as thermal analysis. The weight average molecular weights, $M_w$, glass transition temperatures, $T_g$, and melting temperatures, $T_m$, of both copolymers, as provided by the supplier, are summarized in Table 1. The materials are referred to as HMWPA (high molecular weight PA) and LMWPA (low molecular weight PA) in the presented study. Due to the substantial moisture uptake that is characteristic for polyamides, both materials were dried for 24 h at 80 °C in a Vacutherm VT 6025 vacuum drying oven (Thermo Fisher Scientific, Waltham, MA, USA) before printing or thermal analysis. When printing, the filament was transferred from the vacuum oven to a PrintDry filament drying station (PrintDry, Windsor, ON, Canada), equipped with silica gel. The drying station, set at 70 °C, which is the maximum temperature of the device, allowed the filament to stay dry during printing, since the filament was being fed directly from the dryer into the feeder of the Ultimaker 2 FFF printer (Ultimaker, Geldermalsen, The Netherlands).

**Table 1.** $M_w$, $T_g$ and $T_m$ of the PA 6/66 copolymers.

| Material | $M_w$ [kg/mol] | $T_g$ [°C] | $T_m$ [°C] |
|---|---|---|---|
| HMWPA | 58 | 49 | 199 |
| LMWPA | 24 | 41 | 198 |

A custom G-code was written and uploaded in the Ultimaker 2 FFF printer equipped with a 0.4 mm diameter nozzle to print the required samples. Figure 1 shows the sample geometry, which consisted of a 60 mm wide wall with a thickness of one layer in the y-direction. On top of the brim, defined as Layer 1, 50 layers were added with a layer height of 0.2 mm. Kapton tape was applied on the build plate to ensure sufficient adhesion of the printed sample. During printing, the Ultimaker 2, which is open at the front and top, was covered with the Ultimaker 2 PPMA cover (Ultimaker, Geldermalsen, The Netherlands) to limit the influence of distortions in air flow of the surroundings. However, the temperature of the build environment was not regulated, since the Ultimaker 2 does not allow for environmental control.

Besides using both provided materials, the liquefier temperature, $T_{liquefier}$, build plate temperature, $T_{build\ plate}$, and print speed, $v_{print}$, were varied. Table 2 gives an overview of the different applied print settings.

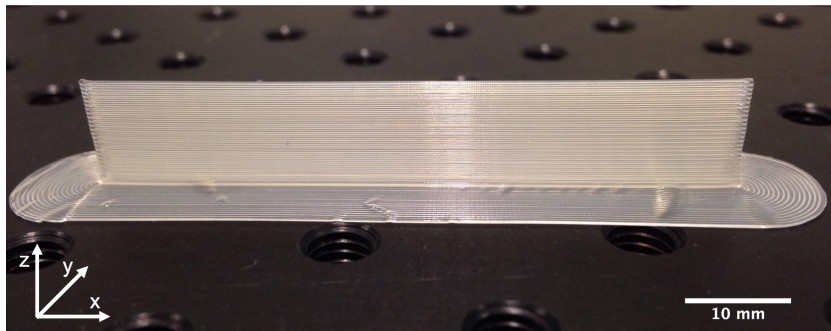

**Figure 1.** The printed single-layer wall geometry.

**Table 2.** An overview of the various print settings for each of the examined conditions.

| Condition | Material | $T_{liquefier}$ [°C] | $T_{build\ plate}$ [°C] | $v_{print}$ [mm/s] |
|:---:|:---:|:---:|:---:|:---:|
| 1 | HMWPA | 260 | 110 | 11 |
| 2 | HMWPA | 260 | 40 | 11 |
| 3 | HMWPA | 240 | 110 | 11 |
| 4 | HMWPA | 260 | 110 | 5.5 |
| 5 | HMWPA | 220 | 110 | 11 |
| 6 | LMWPA | 240 | 110 | 11 |

### 2.2. IR Thermography

During thermal monitoring, an Optris PI 640 IR camera (Optris GmbH, Berlin, Germany) captured the thermal history during the printing process with a resolution of 640 × 480 pixels, at 32 Hz. The measured temperature range was set from 0 °C to 250 °C, with a sensitivity of 75 mK. During printing, the infrared camera resided on the build plate. To shield the IR camera from direct contact with the build plate, the camera was placed on a piece of expanded polystyrene cut to fit the camera, once the brim was printed. Since the internal temperature of the camera cannot exceed 45 °C, the recording time was limited. Hence, the camera could capture the full print as well the cooling down up to several minutes after the print, but not until the printer signals that the print could be removed. The intensity of infrared light with wavelengths between 7.5 and 13 μm was measured and directly converted to temperature by the Optris PIX Connect software, which was used to analyze the temperature variations in the printed samples. For every wall printed at the conditions displayed in Table 2, the two regions of interest were the middle of Layers 10 and 40. To extract temperature profiles from the recorded IR videos, measure areas of 2 × 2 pixels were defined at the regions of interest. The emissivity at the measure areas was set to 0.92, which gave good agreement between the measured and set values of the build plate temperature. The time–temperature data at each measure area could be directly extracted and used as input for the FSC device. Previous literature has suggested a correction for the reflected radiation of the nozzle, thereby also circumventing the need to provide a value for emissivity [36]. However, the software accompanying the IR camera used in this research directly converts infrared intensity to temperature and no direct access to measured intensities is provided by the manufacturer. Therefore, no correction was carried out, and the data were used as provided by the software. It should be noted that the IR camera performs an automatic calibration to prevent excessive thermal drift every 7 s, which results in data loss during approximately 1 s.

### 2.3. Fast Scanning Chip Calorimetry

#### 2.3.1. Device Specifications and Sample Preparation

The Flash DSC 1 (Mettler Toledo, Columbus, OH, USA) was utilized as a tool for simulation of the thermal profiles recorded through IR thermography and assessment of the resulting crystallinity. The Flash DSC 1 is a highly performant FSC device that can reach heating and cooling rates up to

20,000 and 4000 °C/s, respectively, due to the small sample size, which strongly reduces thermal lag. For an extensive description of its working principles and technical specifications, the reader is referred to [49–51]. The required sample mass should lie between 10–20 ng and 1–10 μg. For the measurements in this study, the sample mass, which was not known a priori, was set to 0.1 μg in the device, which had to be corrected for afterwards. These small samples were then applied onto a calorimeter MultiSTAR UFS1 sensor chip (Xensor Integration, Delfgauw, the Netherlands) to perform the measurements. Prior to application of the sample, the sensor chip was conditioned six times to relieve it from thermal stresses. During conditioning, the sensor chip was heated from 45 °C to 450 °C at a rate of 27 °C/s, and then held isothermally at 450 °C for 15 s. In the next step, the UFS1 chip was cooled down from 450 °C to 45 °C at a rate of 100 °C/s, after which it was subsequently heated to 450 °C and again cooled to 45 °C, both at a rate of 100 °C/s. After conditioning, the chip was ready for sample application. Small grains of each material were cut out of the center of the printing filament using a razor blade. With the aid of eyelash tweezers and a microscope, a sample was placed onto the sample area of the chip. The sample was preheated up to 220 °C at 0.5 °C/s, so that it curled up to a more spherical shape and it was fixed onto the chip. Throughout all experiments, nitrogen flowed over the chip at 20 mL/min.

### 2.3.2. Determination of the Critical Heating and Cooling Rate

A critical cooling rate of 250 °C/s was applied, as this proved to suppress crystallization completely, since no visible melting peak could be observed when reheating. This value was validated by earlier work on PA 6 mentioning full amorphization at cooling rates above 200 °C/s [52]. Knowledge of the critical heating rate is essential to suppress cold crystallization and, hence, only measure process-induced crystallization when reheating after the applied temperature profile. To determine this critical heating rate for each material, the sample was first heated at 100 °C/s to 250 °C. The subsequent cycles started by keeping the temperature isothermally at 250 °C for 120 s, and then cooling down to 25 °C at a rate equal to the critical cooling rate and staying isothermal at 25 °C for 10 s. For every cycle, heating from 25 °C to 250 °C was performed at a different heating rate, namely: 5 °C/s, 10 °C/s, 20 °C/s, 50 °C/s and 100 °C/s. A heating rate for which no cold crystallization was observed, was considered to be greater than the critical heating rate.

### 2.3.3. Correction for the Difference in FSC Chip Sample Mass Using DSC

Since the exact sample mass of the polymer samples on the FSC chips was not known, melting enthalpy values obtained from different samples could not be directly compared. Hence, conventional DSC with samples of known mass was used to calculate the FSC sample masses. Aluminum DSC pans were filled with 8.39 and 8.28 mg of HMWPA and LMWPA, respectively, and analyzed in a DSC Q2000 (TA Instruments, New Castle, DE, USA) in a nitrogen atmosphere. The samples were subsequently heated from 20 °C to 250 °C and cooled back down to 20 °C at a rate of 10 °C/min twice. The samples were kept isothermally for 5 min at the temperature limits. To be able to compare the DSC and FSC signals, the FSC samples were subjected to similar thermal cycles with the same cooling rate, but with a heating rate of 100 °C/s, as this gave a stronger signal in the FSC device, allowing for a more accurate melting peak analysis. Since two distinct heating rates were used in the two devices, the heat flow signals [mW or W] should be divided by the respective heating rates [°C/s] to take this into account. The melting peak of each signal was integrated by fitting a baseline onto the normalized heat flow signal [mJ/°C], and integrating the peak area enclosed between the signal and baseline, corresponding to the melting enthalpy [mJ], over temperature. Depending on the shape of the curve of the normalized heat flow signal, it was fitted by either a linear or a quadratic baseline. If a linear scaling between heat flow and sample mass was assumed, the FSC sample mass, $m_{FSC}$, could be directly calculated by Equation (1) using the DSC sample mass, $m_{DSC}$, and the melting enthalpies from DSC, $\Delta H_{m,DSC}$, and FSC, $\Delta H_{m,FSC}$. The integrated heat flow signals acquired from FSC measurements using the thermal profiles associated with different printing conditions, as well as the integrated melting

peaks from the isothermal crystallization measurements, could then be normalized by dividing by the respective sample mass of the used chip. This allowed direct comparison of the results for the HMWPA and LMWPA copolymers. It should be noted that the DSC measurements confirmed the glass transition and melting temperatures of the provided samples.

$$m_{FSC} = \frac{\Delta H_{m,FSC} m_{DSC}}{\Delta H_{m,DSC}} \tag{1}$$

### 2.3.4. Isothermal Crystallization Measurements as a Reference

Since the PA samples are prone to changes, such as sample degradation, post-condensation and spreading, which is especially important when performing a large array of measurements, a correction should be taken into account that rectifies this drift on the resulting data. Therefore, isothermal crystallization measurements were executed for both materials in between every regular thermal profile simulation as they should, in principle, give the same result each time. The protocol for the isothermal crystallization measurement started by heating the sample from 25 °C to 250 °C at 100 °C/s. The sample was then kept at 250 °C for 2 min, after which it was rapidly cooled to 125 °C at 1000 °C/s. To give the material sufficient time for crystallization to occur, it was held isothermally at 125 °C for 5 min. Subsequent heating until 250 °C at 100 °C/s resulted in a melting peak that can be integrated as is specified in Section 2.3.3. Finally, the sample stays at 250 °C for 10 s, after which it was cooled down again to 25 °C at 1000 °C/s. Depending on the amount of measurements that were performed successively, the resulting melting peak data [mJ] were integrated for every three or five isothermal crystallization measurements. A linear relationship could be established when fitting the integrated melting peak data, normalized to the respective sample mass [ng], versus the measurement sequence numbers in the array of consecutively performed measurements. The slope of the produced trend line expresses the drift at which the data either slightly increased or decreased over time. The resulting melting enthalpy value of each measurement, normalized to the sample mass, should therefore be corrected by adding the product of the opposite of the trend line slope and the sequence number of the corresponding measurement to its value.

### 2.3.5. Simulation of IR Thermal Profiles

The Flash DSC 1 is only capable of imposing heating and cooling at a constant rate or isothermal temperature segments. The recorded temperature profiles from IR thermography should thus be linearly approximated for them to be simulated in the FSC device. Careful selection of data points generated sets of temperature segments with constant cooling or heating rate. For each printing condition displayed in Table 2, both the temperature profile for the 10th layer, closer to the build plate, and the 40th layer, farther away from the build plate, were implemented in the Flash DSC 1. As mentioned in Section 2.2, automatic calibration of the infrared camera causes data loss over a period of approximately 1 s. This data loss was compensated for in two ways. At early stages of printing, when large temperature fluctuations occurred, the lost data were replaced by previously captured data from recordings without automatic calibration. This was possible, since thermal drift inherent to the camera without automatic calibration had not yet impacted the recorded temperatures at the early stages of printing. At later stages, heating and cooling rates remained constant over longer time intervals, thus the duration of one linearized segment was much larger than the calibration time of 1 s. Therefore, the linearization was not affected by the automatic calibration. For all recorded temperature profiles, the same protocol was applied, which was comprised of a heating cycle from 25 °C to 250 °C at a rate of 100 °C/s. Secondly, the sample was held isothermally at 250 °C for 120 s to ensure the sample was completely molten. Subsequently, the sample was cooled down from 250 °C at the critical cooling rate of 250 °C/s to the first peak temperature of the simulated temperature profile. After running through the linear approximation of the temperature profile, the sample was heated again from the final temperature of the thermal profile to 250 °C at a rate above the critical heating rate.

The resulting melting peak of this final heating step was used to assess the crystallization. One of the main advantages of using FSC to mimic the thermal history experienced by the deposited layer is the possibility to study the development of crystallinity over the course of printing. The final heating cycle was therefore not only applied at the end of the full recorded temperature profile, but also at distinct points in time during the FFF process. The temperature profiles for all printing conditions were interrupted at three points in time:

1. after the cyclic heating and cooling at the beginning of the temperature profile, resulting from the deposition of new layers;
2. after the print finished, i.e., the last layer was deposited; and
3. at the end of the recorded temperature profile.

For Conditions 1 and 6 in Table 2, the temperature profile for both the 10th and 40th layers were interrupted by a heating cycle after every temperature peak, resulting from the deposition of a new layer, as well as after every linear segment during cooling down of the printed part to monitor the gradual build-up of crystallinity over time in more detail. The resulting heat flow signals [J/s] were normalized by dividing by the applied heating rate [°C/s], since the integrated peak area differs according to the applied rate. A result was obtained in units of heat capacity [J/°C]. The melting peak of each signal was integrated as described in Section 2.3.3. The resulting value of the integrated peak area should then be divided by the respective sample mass of the utilized chip to compare between results obtained for HMWPA and LMWPA. Afterwards, the correction for the inherent drift due to sample changes should be applied. The obtained melting enthalpies were an explicit indication of the degree of crystallinity developed in the printed part. Any difference or trend in melting enthalpy values signified an identical trend in crystallinity. The PA 6 and PA 66 segments of the copolymers were expected to crystallize in their own crystal structure. The values of the melting enthalpy for 100% crystalline PA 6 and PA 66 as reported in the literature amount to 230 and 255.41 J/g, respectively [53]. Thus, if the exact sample composition of each of the copolymers was known, which would be considered identical for both copolymers, a value for the reference melting enthalpy associated with 100% crystallinity could be calculated based on the relative distribution of PA 6 and PA 66 segments. This would allow a direct conversion of the melting enthalpies to absolute crystallinity. Unfortunately, the exact sample composition was unknown. Hence, only melting enthalpies are reported. Earlier research on random PA 6/66 and PA 11/12 copolymers shows a significant influence of comonomer ratio on the observed melting point, which corroborates the assumption of an equal sample composition of the copolymers utilized in this study, since they presented almost identical melting temperatures [54,55].

## 3. Results and Discussion

### 3.1. Thermal Monitoring of the FFF Process

IR thermography directly provides the thermal history of a deposited layer monitored over time. Figure 2 portrays an example of a recorded temperature profile for Layer 10 of Condition 1 with three distinguishable zones. Initially, cyclic heating and cooling can be observed, which is due to deposition of new layers onto the monitored layer, resulting in sharp peaks (Zone 1). Here, the layer experiences high heating and cooling rates up to 200 °C/s within a very narrow time window. Earlier studies reported cooling rates around 400 °C/s and even 700 °C/s for fused deposition of PP and PPS, respectively [17,56]. It should be noted that the recorded peak temperatures are several tens of degrees lower than the set liquefier temperature of 260 °C, as was previously reported in the literature [27,28,33,36]. Other authors have attributed this discrepancy to the limited residence time of the polymer filament in the liquefier, especially at higher print speeds, resulting in a difference of 20 °C between the set liquefier temperature and extrudate temperature at a print speed of 4.5 mm/s, which is still considerably lower than the print speed set at 11 mm/s in this case [57]. Although convection with ambient air is the most dominant cooling mechanism, the conduction from the build

plate becomes visible through the increasing lower limit of the peaks at longer time scales. After deposition of multiple layers, the influence of newly deposited layers loses intensity and the layer temperature cools down to the build plate temperature of 110 °C (Zone 2). Once printing is finished, i.e., the last layer is deposited, the layer temperature cools down even further, yet more gradually, since the build plate itself is cooling down (Zone 3).

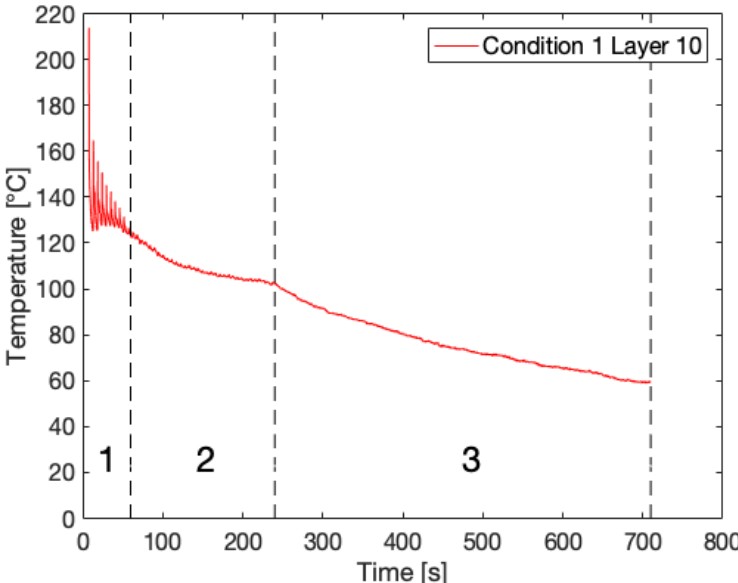

**Figure 2.** An example of a temperature profile for Layer 10 of Condition 1 exhibiting three distinct zones.

An overview of the resulting temperature profiles for the different applied print settings is displayed in Figure 3. All thermal profiles exhibit the three distinct zones. When comparing the temperature profile of the 10th and 40th layers for all conditions, the peak temperatures are similar, whereas the lower limits of the peaks have shifted downwards. This phenomenon can be attributed either to the reduced conduction originating from the build plate at layers printed higher above the build plate or to the decrease in ambient air temperature further away from the build plate, which will enhance convective air cooling, or to a combination of both. The second zone is obviously shorter for Layer 40. The cooling rate in the third zone, when printing has stopped, is also lower for Layer 10 than for Layer 40, since the build plate cools down slowly due to its large thermal inertia. Setting the build plate temperature, $T_{\text{build plate}}$, to 40 °C instead of 110 °C has a significant impact on the temperature experienced by a monitored layer as demonstrated in Figure 3a, comparing Conditions 1 and 2. Although, initially, the first peak temperature lies in the same range, temperature decreases quickly. This is due to the limited conductive heat transfer from the build plate, and the lower ambient temperature. Hence, the overall temperature profiles for Condition 2 are shifted downwards by approximately 50 °C. Figure 3b compares Conditions 1 and 4 and shows the effect of halving the print speed, $v_{\text{print}}$, from 11 mm/s to 5.5 mm/s. As a consequence of reducing the print speed by 50%, the required printing time is doubled. Since the internal temperature of the IR camera cannot exceed 45 °C, it had to be taken off the build plate directly after the print had finished. Therefore, only the first and second zone could be measured for Condition 4. However, the cooling in the third zone is assumed to be similar to Condition 1, as for both conditions the build plate temperature is identical. The temperature peaks at half of the print speed are spread out horizontally, giving more time for a deposited layer to cool down in between deposition of subsequent layers, as is visible from the decrease in the lower peak limits. A slower print speed will increase the residence time of the polymer filament in the heated liquefier as can be observed from the slightly higher peak temperatures recorded for Condition 4. The second zone is also prolonged. A difference in liquefier temperature,

$T_{liquefier}$, of 40 °C has a noticeable effect on the achieved peak temperatures, which are considerably lower at reduced liquefier temperatures, as depicted in Figure 3c. However, due to identical build plate temperatures, the thermal profiles will evolve to similar temperatures over the course of the second and third zone. Finally, the LMWPA and HMPWA printed with identical settings exhibit similar peak temperatures and their recorded temperature profiles almost coincide in the second and third zone. Despite these similarities, the lower peak limits of the HMWPA are remarkably lower compared to the LMWPA, which seems to better retain the heat provided by previously deposited layers, as can be seen in Figure 3d.

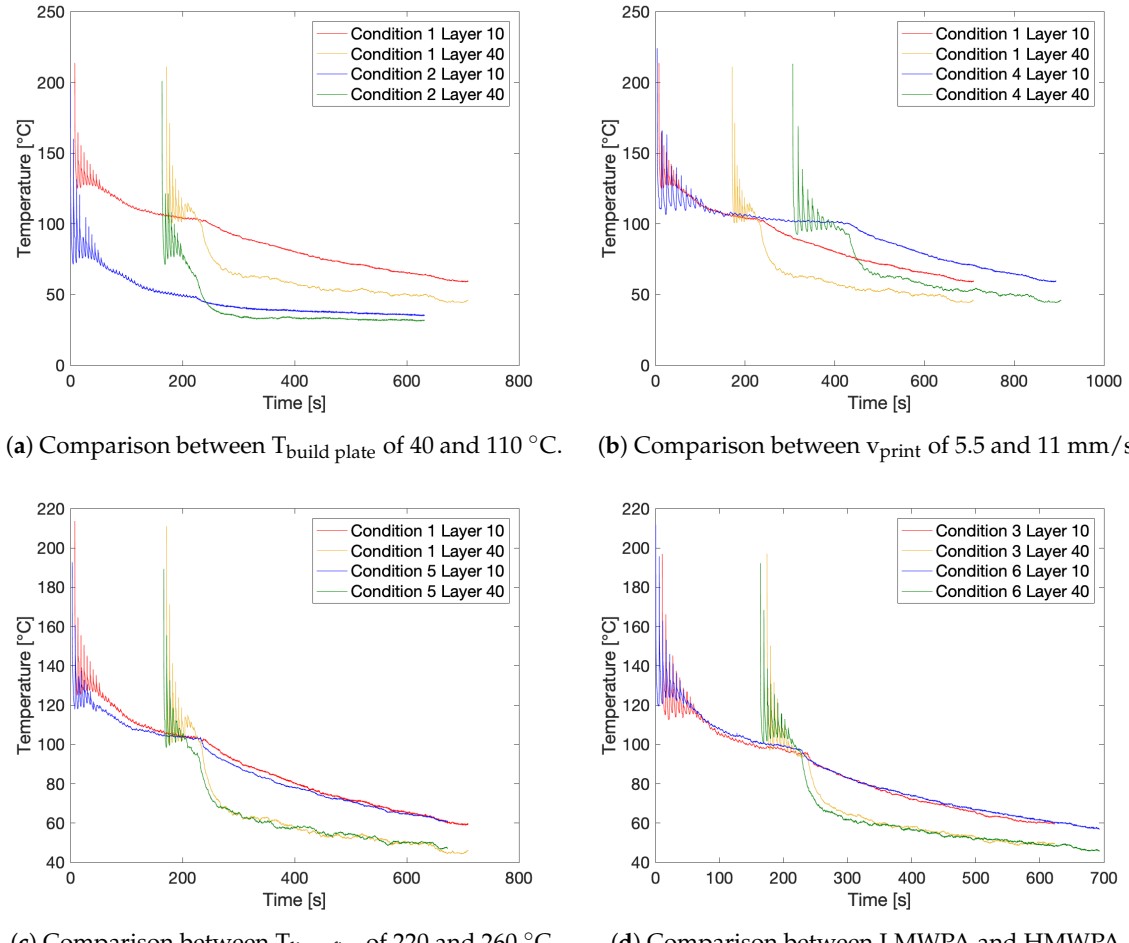

(**a**) Comparison between $T_{build plate}$ of 40 and 110 °C.

(**b**) Comparison between $v_{print}$ of 5.5 and 11 mm/s.

(**c**) Comparison between $T_{liquefier}$ of 220 and 260 °C.

(**d**) Comparison between LMWPA and HMWPA.

**Figure 3.** The recorded thermal profiles for the different printing conditions.

### 3.2. Simulation of Thermal Profiles Using FSC

#### 3.2.1. Critical Heating and Cooling Rate

The heat flow signals [mW] from FSC at increasing heating rates for HMWPA and LMWPA are displayed in Figure 4a,b, respectively. Especially for HMWPA, cold crystallization peaks become clearly visible at heating rates from 5 to 20 °C/s in the range of 50–150 °C. The subsequent melting peaks above 150 °C indicate melting of the formed crystalline structures. Even for a heating rate of 50 °C/s, a very broad cold crystallization peak and successive melting peak could be observed. For LMWPA, these cold crystallization peaks are less apparent. However, a heating rate of 100 °C/s shows no cold crystallization peaks for both materials and is thus adopted as the critical heating rate. For all measurements, a cooling rate of 250 °C/s was applied before heating at the distinct rates. Since no melting peaks emanate when heating at the critical heating rate of 100 °C/s, it is assumed the

materials are completely quenched at a cooling rate of 250 °C/s, thus fully suppressing crystallization, which supports the selection of this rate as the critical cooling rate. Note that the glass transition temperatures of both materials appear at higher cooling rates as a drop in the heat flow signal between 40 and 50 °C.

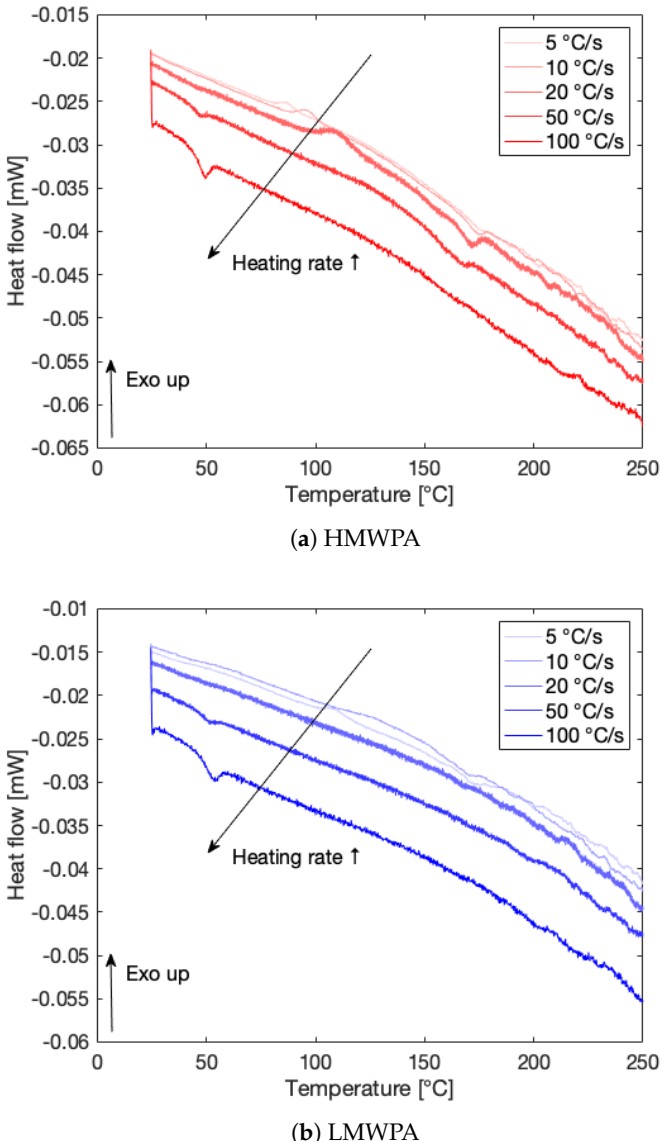

(**a**) HMWPA

(**b**) LMWPA

**Figure 4.** Heat flow signals [mW] from FSC for HMWPA and LMWPA at increasing heating rates to establish the critical heating rate.

### 3.2.2. Correction for Difference in Sample Mass

The melting peaks in the heat flow signals normalized to the heating rates of 100 °C/s and 10 °C/min from FSC [mJ/°C] and DSC [J/°C] can be fitted with a baseline. Figure 5 gives an example of the resulting melting peaks for LMWPA. By integration of the area enclosed by the melting peak and baseline over temperature, a value for the melting enthalpy is obtained. The corresponding FSC sample mass can then be calculated through Equation (1) and amounts to 98.86 and 33.88 ng for the HMWPA and LMWPA FSC samples, respectively. The results for both materials are summarized in Table 3.

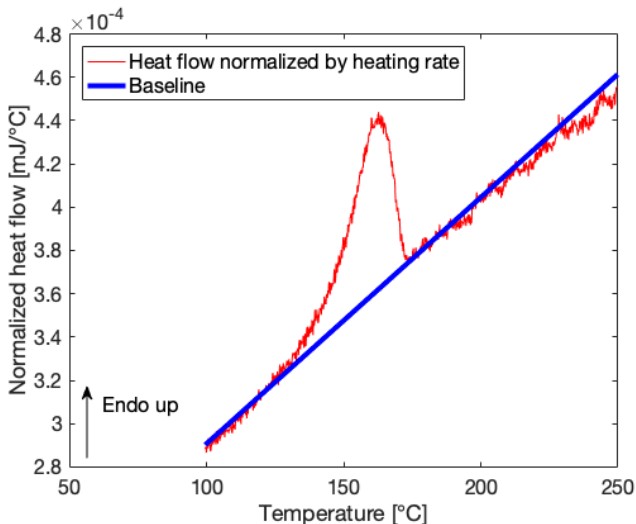

(**a**) Melting peak from FSC and fitted baseline.

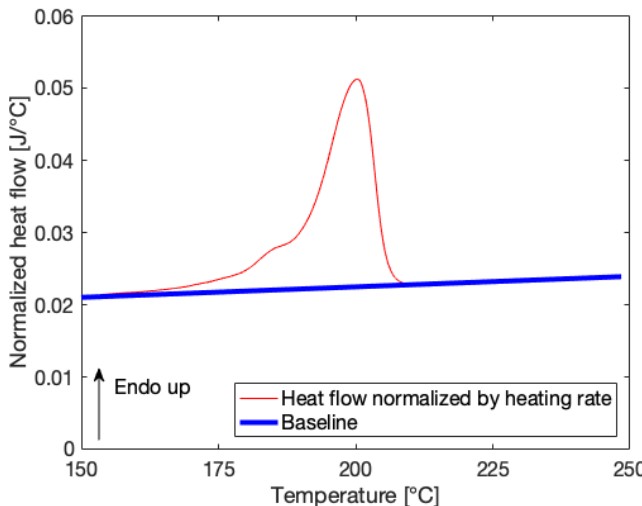

(**b**) Melting peak from DSC and fitted baseline.

**Figure 5.** Heat flow signals normalized by respective heating rates from FSC [mJ/°C] and DSC [J/°C] for LMWPA with baselines fitted for determination of FSC sample chip mass.

**Table 3.** The results of the determination of the FSC sample masses for HMWPA and LMWPA.

| Material | $m_{DSC}$ [mg] | $\Delta H_{m,DSC}$ [J] | $\Delta H_{m,FSC}$ [mJ] | $m_{FSC}$ [ng] |
|---|---|---|---|---|
| HMWPA | 8.39 | 0.37 | $1.53 \cdot 10^{-3}$ | 98.86 |
| LMWPA | 8.28 | 0.34 | $4.00 \cdot 10^{-3}$ | 33.88 |

### 3.2.3. Correction for Drift Due to Sample Changes

For consecutively executed measurements, a correction factor for drift attributed to sample changes was established starting from melting enthalpies normalized to sample mass produced by isothermal crystallization measurements. Melting enthalpies were again obtained by integrating the peak area between the heat flow signal normalized to the heating rate and the baseline fitted onto the heat flow signal. By plotting the melting enthalpies acquired from isothermal crystallization at 125 °C versus the respective sequence number of the measurement in the array of successively performed measurements, a linear trend can be distinguished. Figure 6 illustrates this trend with an example for isothermal crystallization data obtained during FSC measurements corresponding to Conditions 2,

3 and 5, which form one measurement sequence. The opposite of the slope of the linear fit can thus be utilized to correct for the, in this case, upwards data drift inherent to less stable samples such as PA. The product of the correction factor and the respective sequence number of the measurement should then be added to the melting enthalpy value normalized to the sample mass. Three distinct measurement sequences were performed to assess the development of crystallinity for the different printing conditions using FSC. The corresponding correction factors are listed in Table 4.

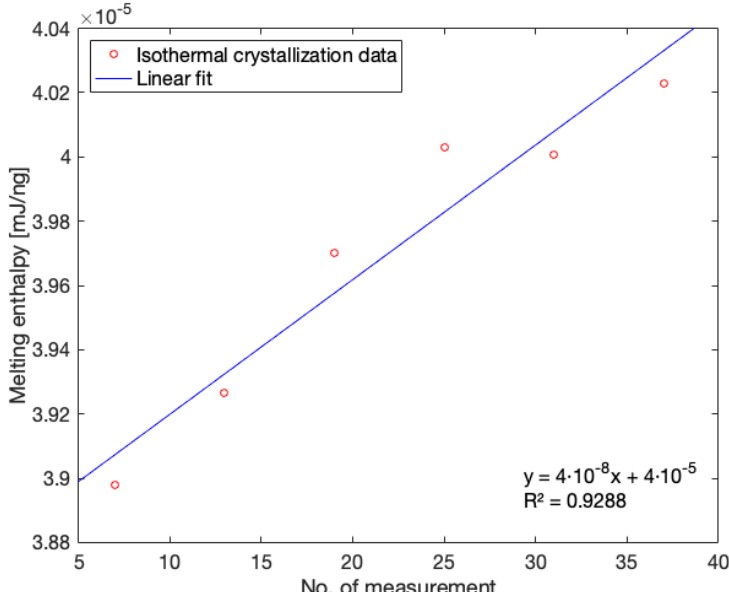

**Figure 6.** The melting enthalpies from isothermal crystallization measurements during FSC measurements of thermal profiles corresponding to Conditions 2, 3 and 5 with linear fit.

**Table 4.** The correction factors for drift due to sample changes for the analyzed print conditions.

| Valid for Condition(s): | Correction Factor |
|---|---|
| 1 & 4 | $1 \cdot 10^{-7}$ |
| 2, 3 & 5 | $-4 \cdot 10^{-8}$ |
| 6 | $7 \cdot 10^{-8}$ |

### 3.2.4. Approximation of Thermal Profiles and Simulation in FSC

To be able to exploit the recorded thermal profiles in the FSC device, the temperature versus time curves should be approximated by linear segments of constant heating or cooling rate. An example of such a linear approximation is depicted in Figure 7 for Condition 2 with markings for the end of each zone in the thermal history. At these points in time, for all analyzed print conditions, the samples were remelted to assess the thus far developed crystallinity through integration of the resulting melting peaks.

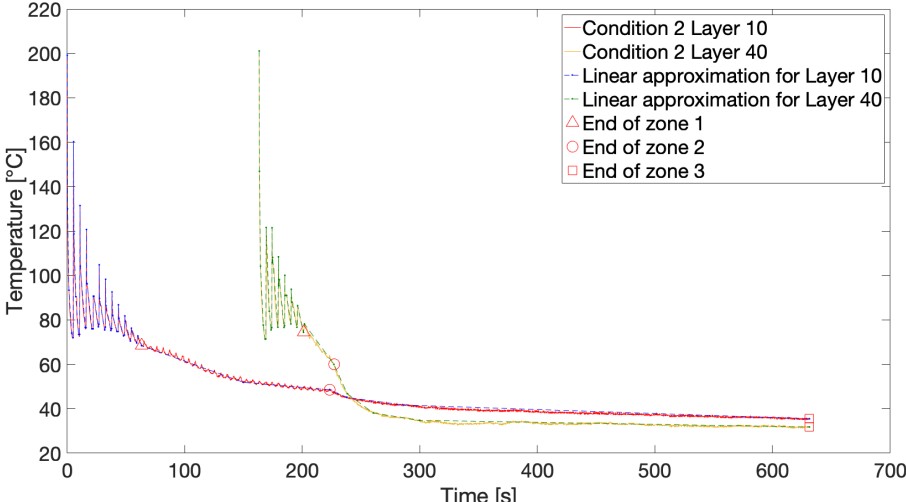

**Figure 7.** The linear approximation of the temperature profiles for Condition 2 with markings for the end of each zone.

Figure 8a shows a more detailed view of the linear approximation imposed onto the thermal profile for Layer 10 of Condition 1. Here, the data loss accompanying the automatic calibration of the IR camera is clearly visible as segments with constant temperature. The linear approximation cuts off these segments, since the period of data loss is smaller than the actual approximated segment of the temperature curve. The peaks of the temperature profile, coinciding with successive deposition of new, hot layers, reveal a dual peak. This dual peak consists of a superposition of a sharp initial peak, attributed to the reflection by the nozzle of the print head, and a second peak with more gradual cooling corresponding to the actual deposition of a new layer. Since no IR intensity values can be accessed through the IR camera software, no correction for the nozzle reflection can be applied on the recorded IR profile as was previously reported in literature [35]. For Conditions 1 and 6, two approximations of their thermal profiles were analyzed to examine the effect of using the corrected linear approximation: one for which the full profile was approximated and one for which correction for the nozzle reflection was applied, shortcutting the first sharp peak, as illustrated in Figure 8b.

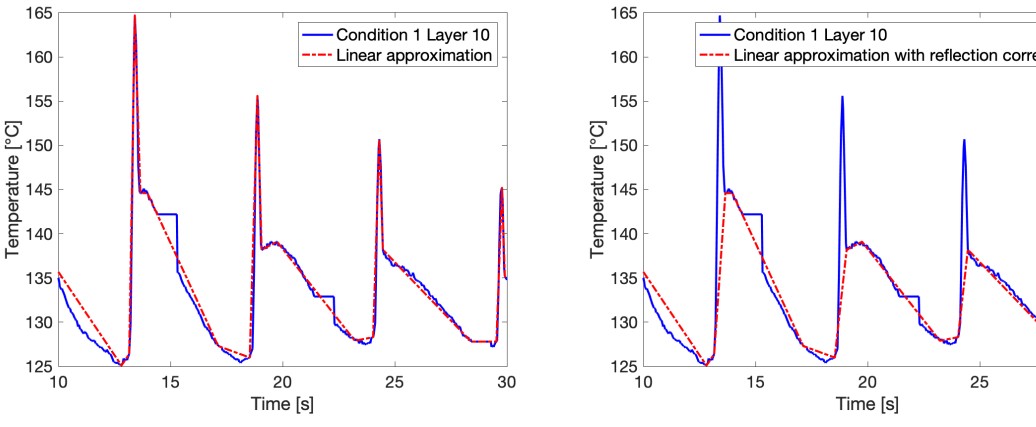

(**a**) Without correction for nozzle reflection.      (**b**) With correction for nozzle reflection.

**Figure 8.** (**a**) A zoomed in view of the linear approximation of the thermal profile, in this case for Layer 10 of Condition 1. (**b**) An example of linear approximation with correction for the peaks associated with reflection from the print head.

The melting enthalpies produced by remelting the FSC samples after each zone in the respective thermal profiles using a full linear approximation without any correction for nozzle reflection are summarized on Figure 9 for all printing conditions. "Low" and "High" refer to the 10th and 40th layers, respectively. A clear increase in melting enthalpy can be observed over the course of printing corresponding to a growing degree of crystallinity of a deposited layer over time. A significant portion of the obtained degree of crystallinity is already acquired during the first zone of cyclic heating and cooling, marking the importance of these cyclic annealing cycles for the development of part crystallinity. For most conditions, the 10th layers exhibit a stronger increase in melting enthalpy over the second zone compared to the 40th layers, as for the latter the second zone is much shorter. When comparing the different conditions using HMWPA, it can be observed that the build plate temperature has a significant impact on the degree of crystallinity that can be obtained during printing. The melting enthalpies reported for Condition 2 with $T_{build\ plate}$ of 40 °C amount to less than 50% of those obtained for Condition 1 printed with $T_{build\ plate}$ of 110 °C. A layer printed closer to the build plate will likewise be more crystalline due to annealing above the material's glass transition temperature through conductive heating from the build plate. This effect is again less pronounced for Condition 2, explained by the fact that the third zone for both monitored layers of Condition 2 almost coincide as a result of the lower build plate temperature in this case, as demonstrated in Figure 3a. For the other conditions, however, there is a clear difference in temperature experienced by the 10th and 40th layers. As a consequence, a gradient in crystallinity is present in the printed wall with crystallinity decreasing when moving away from the build plate. This can again be attributed to a considerably weaker heating of the deposited layers by the build plate, either through conduction or through convective heat transfer by surrounding air heated by the build plate. This suggests the possibility of using an environmentally controlled build chamber, which would surround the printed part and ensure uniform annealing of the deposited layers to mediate the observed gradient. Despite higher peak temperatures associated with a higher liquefier temperature, hardly any change in the degree of crystallinity can be attributed to a higher $T_{liquefier}$. Similarly, halving the print speed has little to no impact on the melting enthalpy values. As a matter of fact, Conditions 1, 3, 4, and 5 possess very similar degrees of crystallinity after the third zone due to the fact that they all experience a similar thermal history during the third zone with a $T_{build\ plate}$ of 110 °C. Findings by other authors support these trends [43–47]. The large difference in melting enthalpy between Condition 6 using LMWPA and the conditions printed with HMWPA illustrates the substantial effect of a decrease in molecular weight on the total degree of crystallinity that is developed over the course of the FFF process, already demonstrated in the literature for PLA [30].

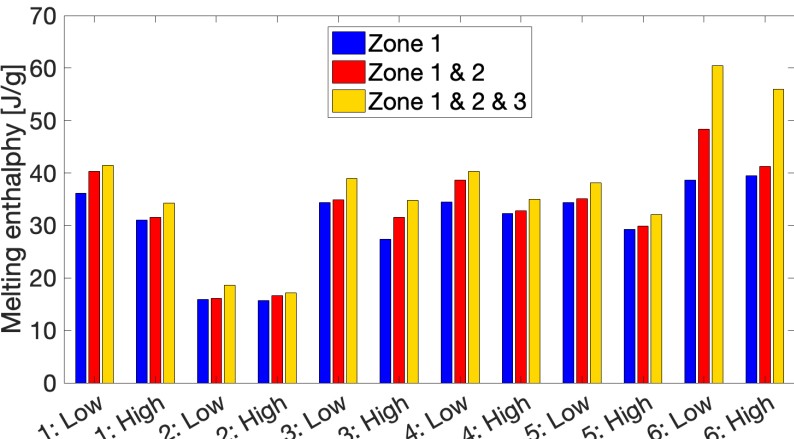

**Figure 9.** An overview of the resulting melting enthalpies after each zone in the thermal profile for the different print conditions.

Applying the correction for reflection originating from the nozzle of the print head onto the simulated thermal profile has very little influence on the eventual melting enthalpies for both Conditions 1 and 6, as can be seen in Figure 10, for both monitored layers. This signifies that, besides the annealing effect of the build plate, the gradual, slower cooling rates after the sharp initial peaks are mainly responsible for the development of crystallinity during the first zone for both conditions. The sharp initial peaks due to reflection from the nozzle will impose significantly high heating and cooling rates over a very short period of time which are insufficient to induce any crystallinity.

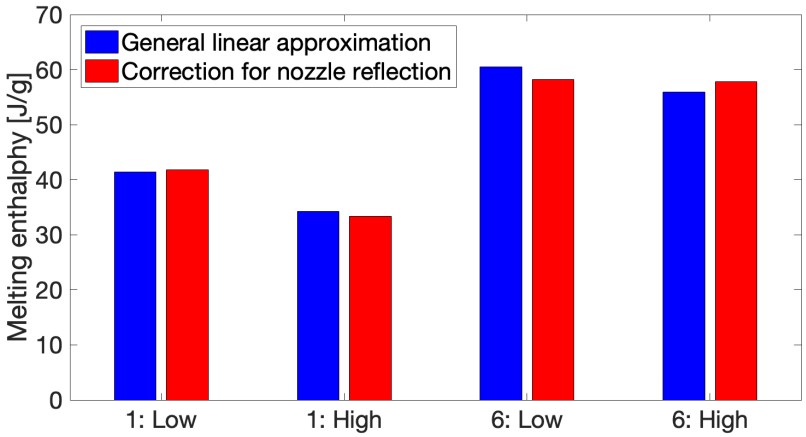

**Figure 10.** The effect of applying the linear approximation corrected for nozzle reflection on the melting enthalpies after simulation of the full thermal profiles.

During simulation of the approximated thermal profiles for Conditions 1 and 6, the FSC samples were remelted after each peak and linear segment in the approximation to get a more in depth picture of the development of crystallinity over time during the FFF process. Figure 11a,b shows the evolution of the resulting melting enthalpies over the course of printing for Conditions 1 and 6, respectively. The corresponding thermal profiles are depicted as well for comparison. It can again be noticed that a considerable portion of the total crystallinity is already acquired during the first zone in the thermal profiles. Besides the evident difference in the total degree of crystallinity after printing between both conditions as mentioned earlier, both samples exhibit a considerably less steep increase in crystallinity after the first two zones. Especially, the HMWPA sample almost reaches a plateau in crystallinity after 300 s of printing. This suggests the material has reached a saturation in crystallinity, meaning that under these conditions, crystallization has died down. The LMWPA copolymer represented by Condition 6 still exhibits a notable increase in crystallinity during the third zone, implying the possibility of crystallization progressing even further if the sample would have remained on the build plate. Hence, the time the part remains on the build plate after printing is crucial to control the eventual crystallinity of the printed part for LMPWA. Although research reported in the literature on PLA mentions a higher degree of crystallinity associated with lower print speed, which could be expected due to the longer printing and thus annealing time, keeping the polymer above its glass transition temperature for a longer time, the results produced in this study do not present these findings [30]. This could be explained by the visible plateau in crystallinity for HMPWA. Despite of the longer printing time for slower print speed, the thermal profiles in Figure 3b suggest a similar cooling cycle for both conditions in the third zone, resulting in a similar plateau value for the crystallinity of both conditions.

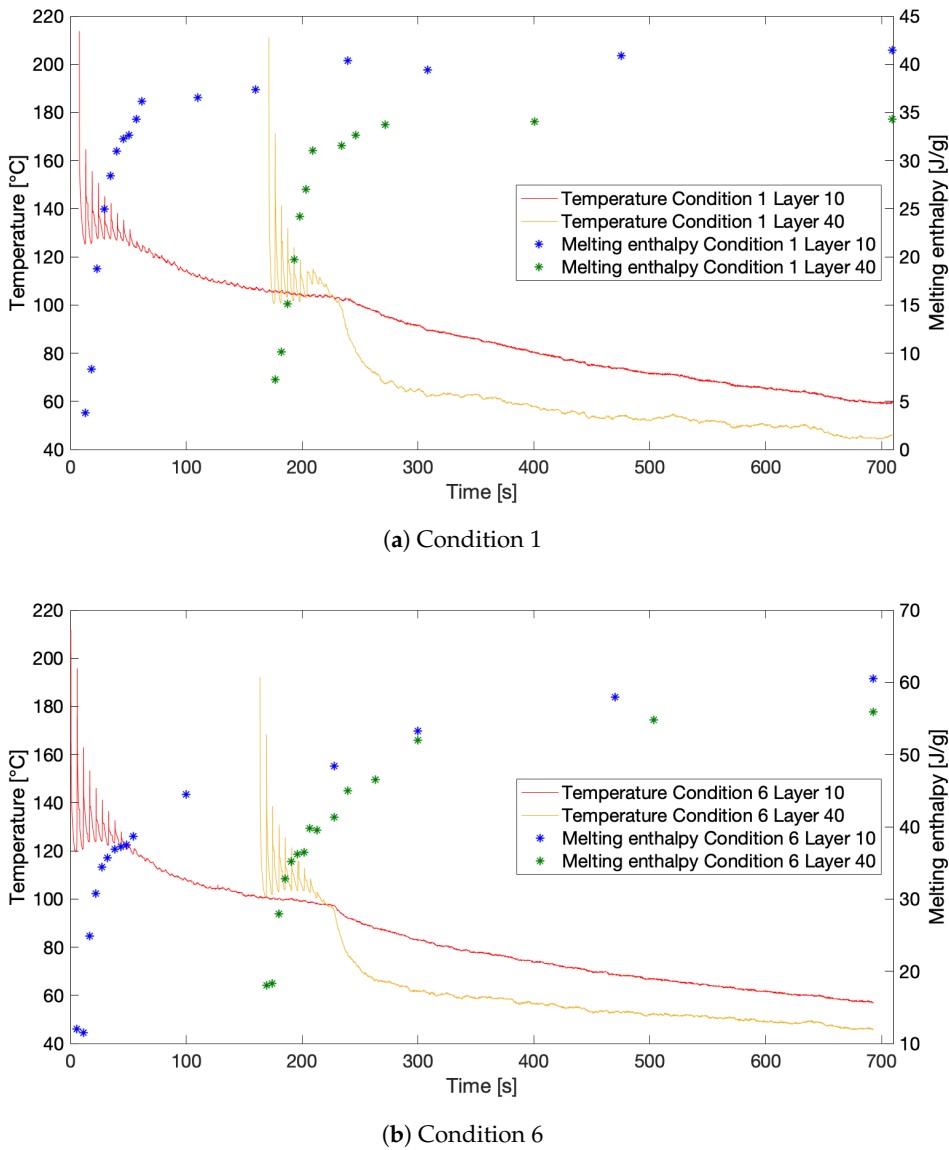

(**a**) Condition 1

(**b**) Condition 6

**Figure 11.** The development of crystallinity over the course of printing compared with the thermal history experienced by the monitored layers for: (**a**) Condition 1 (HMWPA); and (**b**) Condition 6 (LMWPA).

## 4. Conclusions

By utilizing IR thermography to record the thermal history of a deposited layer, the progression of layer temperature over time can be directly extracted and implemented in the FSC device to simulate the associated processing conditions. Several print settings were varied, providing an opportunity to differentiate between the distinct print parameters to determine the most influential settings. Both liquefier temperature and print speed were found to barely impact the developed crystallinity. On the other hand, a substantial effect of the build plate temperature was observed, which can be attributed to the annealing imposed by the build plate. A significant amount of the total crystallinity seemed to have already been amassed during the initial cyclic heating and cooling cycles associated with the successive deposition of new layers in all conditions. However, the molecular weight of the printed polymer seemed to be of key importance for eventual part crystallinity, attaining a considerably higher degree of crystallinity compared to its high molecular weight counterpart printed under identical circumstances. However, the crystallinity of the both copolymers exhibited a notably less steep increase

during the third zone, while the build plate was cooling down. Fast Scanning Chip Calorimetry has thus proven itself as an effective tool to monitor the advancement of crystallinity of a printed part during FFF, which makes it possible to monitor and predict final part crystallinity and performance, once the thermal history of the printed part is established. This even opens up the possibility of employing temperature profiles derived from thermal models of the printing process, which would allow an assessment of final part crystallinity without the need of actually printing a physical object.

**Author Contributions:** Conceptualization, methodology, formal analysis, and investigation, D.V. and M.C.; writing—original draft preparation, D.V.; resources, B.G. and W.Z.; writing—review and editing, B.G., W.Z. and P.V.P.; supervision, W.Z. and P.V.P.; and project administration and funding acquisition, P.V.P.

**Funding:** This research was funded by SIM Flanders (ICON project FLAMINCO; project number: HBC.2017.0325).

**Acknowledgments:** The authors thankfully acknowledge C. Notebaert, K. Van der Flaas and M. Colaers for insightful conversations regarding data analysis and interpretation.

**Conflicts of Interest:** The authors declare no conflict of interest.

## Abbreviations

The following abbreviations are used in this manuscript:

| | |
|---|---|
| AM | Additive Manufacturing |
| 3D | Three-dimensional |
| CAD | Computer-aided-design |
| FFF | Fused Filament Fabrication |
| FDM | Fused Deposition Modeling |
| SLS | Selective Laser Sintering |
| SLA | Stereolithography |
| IR | Infrared |
| DSC | Differential Scanning Calorimetry |
| XRD | X-ray Diffraction |
| $T_g$ | Glass transition temperature [°C] |
| FSC | Fast Scanning Chip Calorimetry |
| $M_w$ | Weight average molecular weight [kg/mol] |
| $T_m$ | Melting temperature [°C] |
| HMWPA | High molecular weight PA |
| LMWPA | Low molecular weight PA |
| $T_{liquefier}$ | Liquefier temperature [°C] |
| $T_{build\ plate}$ | Build plate temperature [°C] |
| $v_{print}$ | Print speed [mm/s] |
| $m_{FSC}$ | FSC sample mass [ng] |
| $m_{DSC}$ | DSC sample mass [mg] |
| $\Delta H_{m,FSC}$ | FSC melting enthalpy [mJ] |
| $\Delta H_{m,DSC}$ | DSC melting enthalpy [J] |

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
