# Peer review of "Assessment of Crystallinity Development during Fused Filament Fabrication through Fast Scanning Chip Calorimetry"

_applsci, doi:10.3390/app9132676_

Round 1

Reviewer 1 Report

An interesting study that many printing with semi-crystalline thermoplastics will find of interest. I have recommended your paper is accepted with minor corrections. A list of corrections is below. Of particular note must be the rewording of the abstract and revision of the Figures listed. 

1. Abstract is very poorly worded and needs to be completely rewritten. The content is fine but the sentence structure is poor and meaning is sometimes lost. 

2. Line 51 - what is reptation? 

3. Line 76 - change to "to enhance crystallinity strongly"

4. Line 88 - change to "mimic the fast heating and cooling rates successfully"

5. Line 155 - change to "suppress crystallization completely"

6.  Line 279 - Loses not looses

7. Figure 3 - each graph should have 4 distinctive colours, it is difficult to distinguish the two similar colours. 

8. Figure 11 - more distinctive colour scheme needed here too. 

Author Response

Response to Reviewer 1 Comments

We thank the reviewer for the positive evaluation of our manuscript.

1.    Abstract is very poorly worded and needs to be completely rewritten. The content is fine but the sentence structure is poor and meaning is sometimes lost.

The abstract will be revised and adjusted as requested.

2.    Line 51 - what is reptation? 

Reptation refers to the motion of polymer chains in an entangled poiymer melt. This has been described by Doi and Edwards in their so-called ‘reptation model’. This model forms the basis to describe molecular diffusion of polymer chains. We believe that this should be common knowledge in the field of polymer physics and opted not to clarify this issue.

3.    Line 76 - change to "to enhance crystallinity strongly"

The change will be implemented as requested.

4.    Line 88 - change to "mimic the fast heating and cooling rates successfully"

The change will be implemented as requested.

5.    Line 155 - change to "suppress crystallization completely”

The change will be implemented as requested.

6.    Line 279 - Loses not looses

The change will be implemented as requested.

7.    Figure 3 - each graph should have 4 distinctive colours, it is difficult to distinguish the two similar colours.

The change will be implemented as requested.

8.    Figure 11 - more distinctive colour scheme needed here too

The change will be implemented as requested.

Reviewer 2 Report

This paper examines the effects of the printing parameters on the crystallinity in the printed parts. This is a very inspiring manuscript since nowadays more and more functional engineering parts are manufactured from FFF style 3D printing; the crystallinity in the parts is critical to the final properties and their reliability. This area has not been intensively investigated, and this manuscript had a good start in this area. However, there are some comments below that need to be addressed.

1.       In addition to the printing parameters, the environment conditions during the print should be mentioned, e.g. the printing was done in a closed environment or an open place with high air-circulation rate, the temperature of the air, etc.

2.       The authors did a lot of work on applying assumptions and corrections to simulate printing temperature profile in the FSC and get the crystallinity results as accurately as possible. While these assumptions and corrections seem reasonable, could the authors also run a sample or two from the actual printed single-layer wall (described in the section 2.1) on the DSC to get the actual crystallinity results and compare to those from FSC under simulated conditions? That could make the assumptions and corrections more convincing.

3.       While the authors have done a good amount of job on getting the enthalpy measurements under simulated conditions on the FSC, no mechanical property was tested in this study. Whether and how exactly the crystallinity could affect the mechanical properties of their PA printed parts are the base of this manuscript: if there is no effects on the mechanical properties, no need to investigate more; if there is, is there an optimal crystallinity to benefit the properties? Some work needs to be done in the evaluation of the mechanical property test.

4.       The authors mentioned the ‘co-crystallization across the interface’, assuming the ‘interface’ means between the layers, but all the FSC measurements were done for individual layers. How did the FSC results relate to the ‘co-crystallization across the interface’?

5.       Does the conclusion in the manuscript apply to all the semi-crystalline polymers, or only to the PA 6/66 random copolymers used in the experiment? PA 6/66 has hydrogen bonding within the polymer network, while some other semi-crystalline polymers do not, such as polypropylene, polyethylene. Will the conclusion still hold up for these polymers? If not, the authors would need to state that in the conclusion.

In summary, this is an original and interesting manuscript that merits publication once the above concerns/comments get addressed.

Author Response

Response to Reviewer 2 Comments

We thank the reviewer for the positive evaluation of our manuscript.  Please find below the answers to the issues you raised.  

1.    In addition to the printing parameters, the environment conditions during the print should be mentioned, e.g. the printing was done in a closed environment or an open place with high air-circulation rate, the temperature of the air, etc.

This is a valid point. The Ultimaker 2 was covered during printing with a plexiglass cover which can be bought as an addition to the print set-up by Ultimaker to limit the influence of distortions in air flow in the room, since the Ultimaker 2 is an open printer. However, no environmental temperature has been recorded, since no additional environmental control is applied as this is not possible with the Ultimaker 2. We will add this in Section 2.1. 

2.    The authors did a lot of work on applying assumptions and corrections to simulate printing temperature profile in the FSC and get the crystallinity results as accurately as possible. While these assumptions and corrections seem reasonable, could the authors also run a sample or two from the actual printed single-layer wall (described in the section 2.1) on the DSC to get the actual crystallinity results and compare to those from FSC under simulated conditions? That could make the assumptions and corrections more convincing.

We thank the reviewer for this comment. DSC requires a minimum sample mass to be able to accurately perform thermal analysis. To apply this on the single-layer wall, a sample should be cut out from the wall which would automatically comprise of multiple layers to meet the sample mass requirement. And hence, crystallinity is not measured locally as we do now with FSC. In-situ XRD-measurements during printing will be used in a follow-up study to validate the findings from FSC.  This however was not the scope of this paper since we wanted to demonstrate the use of FSC as an important tool to follow crystallization during printing. 

3.    While the authors have done a good amount of job on getting the enthalpy measurements under simulated conditions on the FSC, no mechanical property was tested in this study. Whether and how exactly the crystallinity could affect the mechanical properties of their PA printed parts are the base of this manuscript: if there is no effects on the mechanical properties, no need to investigate more; if there is, is there an optimal crystallinity to benefit the properties? Some work needs to be done in the evaluation of the mechanical property test.

As was pointed out above, this manuscript is mainly focussed on the methodology of using FSC and IR thermography as a tool to monitor crystallinity development during FFF printing. Future work will look into the effect of crystallinity on layer adhesion through mechanical tests. 

4.    The authors mentioned the ‘co-crystallization across the interface’, assuming the ‘interface’ means between the layers, but all the FSC measurements were done for individual layers. How did the FSC results relate to the ‘co-crystallization across the interface’?

The interface indeed refers to the interface in between successively printed layers. Currently, the FSC results cannot be used to validate co-crystallization. The effect of co-crystallization was merely mentioned in the introduction to illustrate the possibility of crystallization having a possible positive effect on layer adhesion, since it is often described as being unfavourable for welding since it would hinder molecular diffusion.

5.    Does the conclusion in the manuscript apply to all the semi-crystalline polymers, or only to the PA 6/66 random copolymers used in the experiment? PA 6/66 has hydrogen bonding within the polymer network, while some other semi-crystalline polymers do not, such as polypropylene, polyethylene. Will the conclusion still hold up for these polymers? If not, the authors would need to state that in the conclusion.

Other semi-crystalline polymers will be most likely printed at other specific print settings and will exhibit different crystallization kinetics. Hence, the extent of the observed trends could differ when comparing various semi-crystallinity polymers. However, the trends found in this manuscript for the PA 6/66 random copolymers correspond to earlier work on PLA by Srinivas et al. as was mentioned in the paper, which validates the obvious influences of the build plate temperature and molecular weight on the attained crystallinity. 

Reviewer 3 Report

The authors should be congratulated on an excellent research paper. Crystallization is an underexplored area in FFF / FDM.

I only have two comments (and to clarify, I am happy with the paper, I would just like to see if the authors think it appropriate to consider the following areas);

1) with regards to the drying of the material, the authors are correct to dry the materials before printing but there are two areas I believe they should make reference to;

(A) how long does the print take - do the authors believe the filament could take up moisture during printing (I.e could material printed initially have a lower moisture content than material printed near the end of the print?) And if so could that affect the crystallization kinetics 

(B) overdrying of resins is a common polymer processing danger. It can occur by drying a rein for too long or by repeatedly drying the same batch of resin. It can drive off some low mw additives or oligimers etc - do the authors think that repeated drying of the filament over several prints may alter their findings? Or is this less of a risk than with traditional pellets due to the difference in surface area between pellets and filament?

2) in terms of Crystallinity, I believe it would have been useful to utilize XRD to measure the samples to supplement the work. 

Author Response

Response to Reviewer 3 Comments

We thank the reviewer for the nice comments.  Below, we answer the issues you raised.

1) with regards to the drying of the material, the authors are correct to dry the materials before printing but there are two areas I believe they should make reference to;

(A)  how long does the print take - do the authors believe the filament could take up moisture during printing (I.e could material printed initially have a lower moisture content than material printed near the end of the print?) And if so could that affect the crystallization kinetics

We agree with the reviewer that moisture is an issue when dealing with materials like PA.

The printing filament is placed in a PrintDry filament dryer which is set at 70 °C (this is the maximum temperature) and equipped with silica gel immediately after it has been taken out of the vacuum oven. This device allows for the filament to stay dry during printing, since the filament is being fed directly from the dryer into the feeder of the Ultimaker 2. We will clarify this in Section 2.1. 

(B)overdrying of resins is a common polymer processing danger. It can occur by drying a resin for too long or by repeatedly drying the same batch of resin. It can drive off some low mw additives or oligimers etc - do the authors think that repeated drying of the filament over several prints may alter their findings? Or is this less of a risk than with traditional pellets due to the difference in surface area between pellets and filament? 

The drying procedure has been suggested by the supplier and no significant effect of drying has been found. Indeed, due to a difference in surface area the filament would be less prone to overdrying as compared to pellets.

2) in terms of Crystallinity, I believe it would have been useful to utilize XRD to measure the samples to supplement the work.

This is a valid argument. This manuscript is focussed on the methodology of using FSC combined with IR thermography to study crystallinity development.  However, future work will be concentrated on validating FSC results with XRD combined with mechanical tests to check possible influence of crystallinity on weld strength. 

Round 2

Reviewer 2 Report

The authors have addressed all the concerns stated in the reviewers' reports in the revision. It would be much more helpful to the audience if the authors could also include their future work after the conclusion part. But in general, this manuscript is suitable to publish.